# Microscopic distance from tumor invasion front to serosa might be a useful predictive factor for peritoneal recurrence after curative resection of T3-gastric cancer

Shingo Togano[1,2], Masakazu Yashiro[1,2,3]*, Yuichiro Miki[1,2], Yurie Yamamoto[2,3], Tomohiro Sera[1,2], Yukako Kushitani[1,2], Atsushi Sugimoto[1,2], Shuhei Kushiyama[1,2], Sadaaki Nishimura[1,2], Kenji Kuroda[1,2], Tomohisa Okuno[1,2], Mami Yoshii[1], Tatsuro Tamura[1], Takahiro Toyokawa[1], Hiroaki Tanaka[1], Kazuya Muguruma[1], Sayaka Tanaka[4], Masaichi Ohira[1]

1 Department of Gastroenterological Surgery, Osaka City University Graduate School of Medicine, Osaka, Japan, 2 Molecular Oncology and Therapeutics, Osaka City University Graduate School of Medicine, Osaka, Japan, 3 Cancer Center for Translational Research, Osaka City University Graduate School of Medicine, Osaka, Japan, 4 Department of Diagnostic Pathology, Osaka City University Graduate School of Medicine, Osaka, Japan

* m9312510@med.osaka-cu.ac.jp

## Abstract

### Background

Peritoneal recurrence is one of the most frequent recurrent diseases in gastric cancer. Although the exposure of cancer cells to the serosal surface is considered a common risk factor for peritoneal recurrence, there are some cases of peritoneal recurrence without infiltration to the serosal surface even after curative surgery. This study sought to clarify the risk factors of peritoneal recurrence in the absence of invasion to the serosal surface.

### Materials and methods

Ninety-six patients with gastric cancer who underwent curative surgery were enrolled. In all 96 cases, the depth of tumor invasion was subserosal (T3). The microscopic distance from the tumor invasion front to the serosa (DIFS) was measured using tissue slides by H&E staining and pan-cytokeratin staining. E-cadherin expression was evaluated by immunohistochemical staining.

### Results

Among the 96 patients, 16 developed peritoneal recurrence after curative surgery. The DIFS of the tumors with peritoneal recurrence (156±220 μm) was significantly shorter (p = 0.011) than that without peritoneal recurrence (360±478 μm). Peritoneal recurrence was significantly correlated with DIFS ≤234 μm (p = 0.023), but not with E-cadherin expression. The prognosis of DIFS ≤234 μm was significantly poorer than that of DIFS >234 μm (log rank, p = 0.007). A multivariate analysis of the patients' five-year overall survival revealed

**Funding:** This work was supported by KAKENHI Grant-in-Aid for Scientific Research, Nos. 18H02883(M.Y.).

**Competing interests:** The authors have declared that no competing interests exist.

that DIFS ≤234 μm and lymph node metastasis were significantly correlated with survival (p = 0.005, p = 0.032, respectively).

## Conclusion

The measurement of the DIFS might be useful for the prediction of peritoneal recurrence in T3-gastric cancer patients after curative surgery.

## Introduction

Among all malignant neoplasms worldwide, gastric cancer ranks fifth for cancer incidence and second for cancer deaths [1]. Although curative resection (R0) with lymph node dissection plus adjuvant chemotherapy has prolonged the survival of patients with gastric cancer, the recurrence rate of R0 cases remains around 30% in patients at stage II/III [2, 3]. Peritoneal recurrence is the most frequent recurrence pattern in patients with gastric cancer after curative resection, and as such, peritoneal recurrence is the most common cause of subsequent cancer death [4–7].

The exposure of cancer cells to the serosal surface (i.e., T4) is a common risk factor for and accounts for most cases of peritoneal recurrence [8, 9]. However, peritoneal recurrence can develop in not only T4 cases but also cases without the exposure of cancer cells to the serosal surface (i.e., T3). According to the Japanese Research Society for Gastric Cancer, peritoneal recurrence was the cause of death in 2.3% of T1 cases, 6.9% of T2 cases, 17.2% of T3 cases, 33.4% of T4 cases of gastric cancer[9].

It has been reported that E-cadherin is one of important factors for tumor invasion and distant metastasis in some solid cancers[8, 10–12]. Taken together, we previously reported the correlation between the microscopic distance from the tumor invasion front to the serosa (DIFS) and serosal exposure of gastric cancer cells, and speculated that DIFS might be associated with peritoneal recurrence[3]. Then, in this study we focused on the significance of DIFS and E-cadherin in peritoneal recurrence.

The present study was conducted to clarify the risk factors of peritoneal recurrence after R0 surgery for T3-stage gastric cancer.

## Materials and methods

### Patients

A total of Ninety-six patients with gastric cancer, who received gastrectomy between 2000 and 2016 at Osaka City University, were enrolled in this study. The inclusion criteria were as follows; 1. histologically proven gastric adenocarcinoma; 2. the depth of tumor invasion was T3; 3. curative operation; 4. intraoperative peritoneal lavage cytology-negative (Fig 1). Since the peritoneal recurrence of T1 and T2 cancers has been considered to develop via trans-lymphatic pathway[13, 14], we excluded T1 and T2 cases in this study. The follow-up period was 60 months, and the median follow-up was 49.3 months. The follow-up program of postoperative surveillance consisted of computed tomography, and ultrasound performed every 3 months in order to diagnose recurrent diseases.

The pathological data was recorded according to the eighth edition of TNM Classification [15]. Pathologic examination was performed using the section which include center of the tumor. Macroscopic type were determined according to the Japanese Gastric Cancer

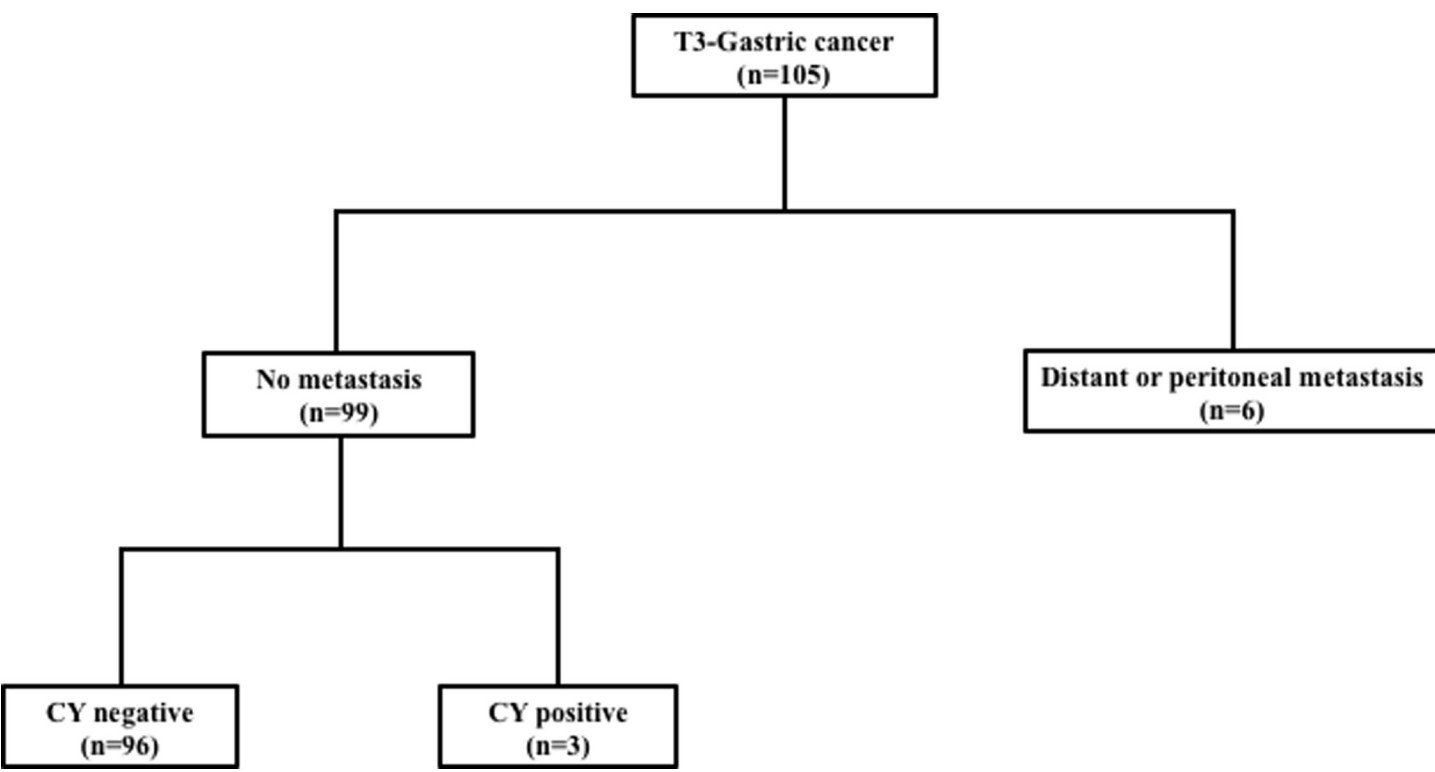

**Fig 1. The inclusion criteria in flowchart.** The inclusion criteria were as follows; 1. histologically proven gastric adenocarcinoma; 2. the depth of tumor invasion was T3; 3. curative operation; 4. intraoperative peritoneal lavage cytology-negative (Fig 1).

Association classification with third English edition[16]. This study was approved by the Osaka City University Ethics Committee (approval number 924). Written informed consent for research was obtained from patients.

## Immunohistochemical techniques

After gastrectomy, the gastric tumor was immediately treated with 10% formalin neutral buffer solution for 24–72 hours. Paraffin-embedded sections were de-paraffinized in xylene and dehydrated through graded ethanol. The sections were heated for 10 min at 105˚C by autoclave in Target Retrieval Solution (DAKO, Carpinteria, CA, USA). Then sections were incubated with 3% hydrogen peroxide to block endogenous peroxidase activity before immunohistochemistry using the following antibodies: anti pan-cytokeratin (26411-1-AP, 1:2000; Proteintech, Rosemont, IL, USA) and anti E-cadherin (NCH-38, 1:100; DAKO). The specimens were incubated with E-cadherin and pan-cytokeratin antibody for overnight at 4˚C. The sections were incubated with an appropriate immunoglobulin G for 10 min, followed by three washes with phosphate-buffered saline (PBS). The slides were treated with streptavidin-peroxidase reagent, and were incubated in PBS with diaminobenzidine and 1% (vol/vol) hydrogen peroxide, followed by counterstaining with Mayer's hematoxylin and subsequently examined using light microscopy.

## Measurement of the microscopic distance from tumor invasion front to serosa (DIFS)

Pan-cytokeratin staining, that are detectable epithelial components including cancer cells, was used to identify the tumor invasion front. We checked invasion depth of all specimens of the tumor using H-E staining slides. After selecting three specimens with high invasion depth, the three specimens were stained by pan-cytokeratin and measured DIFS of 3 slides. The shortest distance was defined as DIFS of the case. DIFS was measured using the microscope (BZ-X710, Keyence, Osaka, Japan).

## Evaluation of E-cadherin expression

E-cadherin expression was evaluated by intensity of staining and percentage of stained tumor cells at the invading tumor front: intensity was given scores 0–3 (0 = no, 1 = weak, 2 = moderate, 3 = intense), and frequency of positive cells was determined 0–24%, 25–49%, 50–74%, 75–100%. Expressions were considered positive when intensity scores ≥2 and frequency ≥50%, and negative when intensity scores ≤1 or frequency ≤49%. The pathologist, Dr Sayaka Tanaka, checked DIFS and E-cadherin expression. Dr Togano S, Dr Yashiro M, and Dr Tanaka S checked DIFS and E-cadherin expression.

## Statistical analysis

The chi-square test was used to determine the significance of the differences between the covariates. The durations from surgery to peritoneal recurrence were estimated by the Kaplan-Meier method and compared using the log-rank test. Multivariate analysis with respect to peritoneal recurrence was performed using logistic regression analysis. Multivariate analysis with respect to five-year overall survival was performed using Cox proportional hazard model. Covariates were selected from those with significant differences in univariate analysis. All statistical analyses were performed using the JMP statistical software (version 13.2; SAS Institute, Cary, NC). Two-sided probability P values of $< 0.05$ were considered to be statistically significant.

# Results

## Correlations between peritoneal recurrence and clinicopathologic features

Post-operative recurrence was confirmed at the peritoneum in 16 of the 96 cases. Five of the 16 peritoneal recurrence cases developed liver recurrence, and two developed lymph node recurrence. Tables 1 and S1 shows the correlations between peritoneal recurrence and clinicopathologic features. There was a significant correlation between peritoneal recurrence and lymph node metastasis (p = 0.012).

## Correlations between peritoneal recurrence and the DIFS or E-cadherin expression

Pan-cytokeratin was expressed mainly in the cell membrane of the cancer cells (Fig 2). The DIFS of the tumors with peritoneal recurrence (156±220 μm, mean±std. dev.) was significantly shorter than that of the tumors without peritoneal recurrence (360±478 μm) (p = 0.011, t-test). The cutoff value for the DIFS was determined as 234 μm based on the results of the receiver operating characteristics (ROC) curve (Fig 3).

E-cadherin was mainly expressed in the cell membrane of cancer cells (Fig 4). Table 1 shows the correlations between peritoneal recurrence and the DIFS or E-cadherin expression.

**Table 1. Correlation between peritoneal recurrence and clinicopathologic features in 96 cases at T3 stage.**

| Clinicopathologic features | Peritoneal recurrence (n = 16) | No peritoneal recurrence (n = 80) | p value |
|---|---|---|---|
| Age | | | |
| < 70 Years | 10 (16.9%) | 49 (83.1%) | |
| ≥ 70 Years | 6 (16.2%) | 31 (83.8%) | 0.925 |
| Sex | | | |
| Male | 6 (19.4%) | 25 (80.6%) | |
| Female | 10 (15.4%) | 55 (74.6%) | 0.626 |
| Macroscopic type[a] | | | |
| type1-2 | 7 (18.4%) | 31 (81.6%) | |
| type3-4 | 9 (15.5%) | 49 (84.5%) | 0.709 |
| Histological type | | | |
| intestinal | 9 (15.5%) | 49 (84.5%) | |
| diffuse | 7 (18.4%) | 31 (81.6%) | 0.709 |
| LN metastasis[b] | | | |
| negative | 3 (6.3%) | 45 (93.7%) | |
| positive | 13 (27.1%) | 35 (72.9%) | 0.012 |
| INF[c] | | | |
| a/b | 9 (12.9%) | 61 (87.1%) | |
| c | 7 (30.4%) | 16 (69.6%) | 0.064 |
| Lymphatic invasion | | | |
| negative | 2 (7.7%) | 24 (92.3%) | |
| positive | 14 (20.0%) | 56 (80.0%) | 0.221 |
| Vascular invasion | | | |
| negative | 14 (18.9%) | 60 (81.1%) | |
| positive | 2 (9.1%) | 20 (90.9%) | 0.278 |
| Tumor size | | | |
| < 50 mm | 7 (14.3%) | 42 (85.7%) | |
| ≥ 50 mm | 9 (19.1%) | 38 (80.9%) | 0.523 |
| DIFS[d] | | | |
| ≤ 234 μm | 14 (17.2%) | 44 (82.8%) | |
| > 234 μm | 2 (5.3%) | 36 (94.7%) | 0.023 |
| E-cadherin | | | |
| negative | 7 (24.1%) | 22 (75.9%) | |
| positive | 2 (13.4%) | 36 (86.6%) | 0.196 |

[a]: Macroscopic type; The classification according to the general rules for gastric cancer study of the Japanese Research Society for Gastric Cancer

[b]: LN metastasis; Lymph node metastasis

[c]: INF; Pattern of tumor infiltration into the surrounding tissue. The predominant pattern of infiltrating growth into the surrounding tissue is classified as follows; INF a: The tumor shows expanding growth and a distinct border with the surrounding tissue. INF b: This category is between INF a and INF c. INF c: The tumor shows infiltrating growth and an indistinct border with the surrounding tissue.

[d]: DIFS; the microscopic distance from tumor invasion front to serosa

Peritoneal recurrence was significantly correlated with DIFS ≤234 μm (p = 0.023), but not with E-cadherin expression.

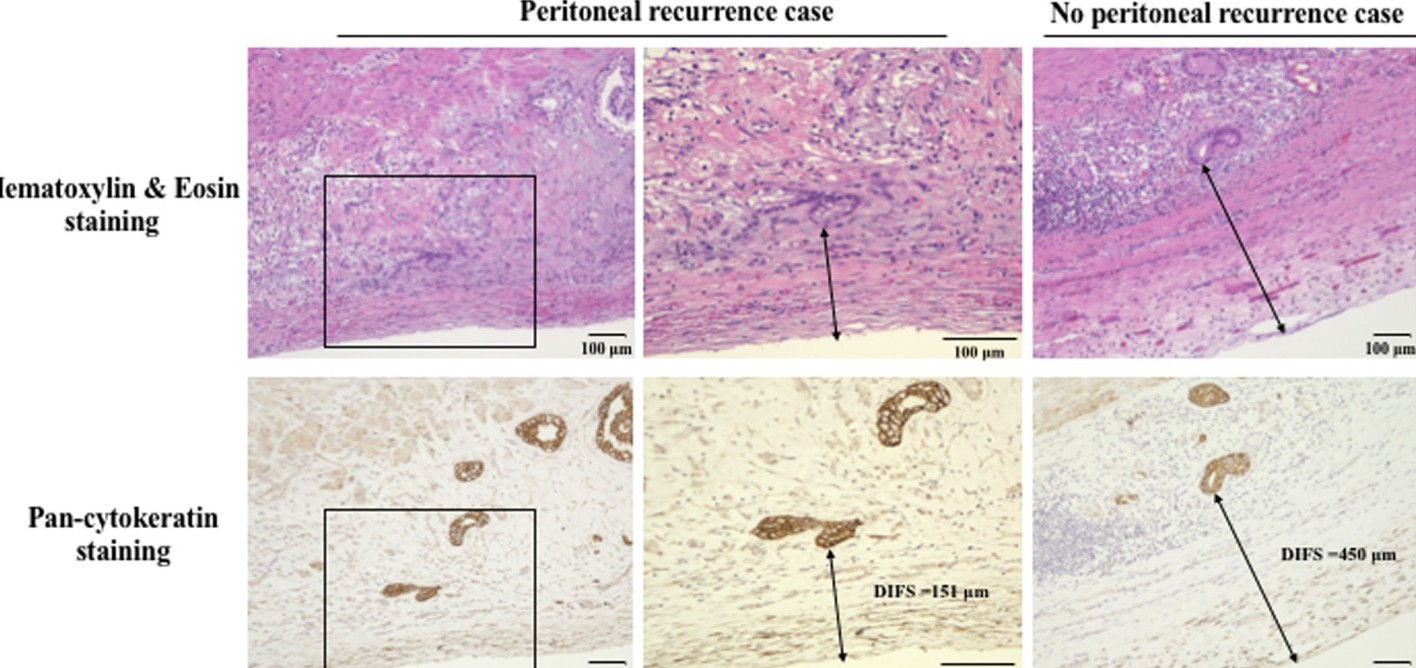

**Fig 2. The microscopic distance from the tumor invasion front to the serosa.** The microscopic distance from the tumor invasion front to the serosa (DIFS) was calculated by H&E staining and/or pan-cytokeratin staining. Pan-cytokeratin staining was used to determine the cancer cells at the invasion front.

## Risk factors of peritoneal recurrence

Table 2 summarizes the results of the univariate and multivariate analyses with respect to peritoneal recurrence. A DIFS ≤234 μm and lymph node metastasis were independent risk factors for peritoneal recurrence (p = 0.049, p = 0.023, respectively).

## Survival

Fig 5 provides the Kaplan-Meier survival curve for the 96 patients. The prognosis of the patients with a DIFS ≤234 μm was significantly poorer than that of the patients with a DIFS >234 μm (log rank, p = 0.007). The prognosis of the lymph node metastasis (N1) patients was significantly poorer than that of the patients without lymph node metastasis (N0) (log rank, p = 0.017). In contrast, no significant correlation was found between E-cadherin expression and prognosis. A multivariate analysis with respect to five-year overall survival revealed that DIFS ≤234 μm and lymph node metastasis were significantly (p = 0.005, and p = 0.032, respectively) correlated with survival (Table 3).

## Discussion

The DIFS was associated with peritoneal recurrence. The cut-off value of DIFS was determined as 234 μm in accordance with the ROC curve analysis. In the multivariate analysis, DIFS ≤234 μm was an independent risk factor for peritoneal recurrence. These findings suggest that the DIFS is an important pathologic factor that could be used to predict the prognosis of patients with T3-stage gastric cancer.

According to guideline, tegafur-gimeracil-oteracil (S-1) monotherapy is recommended for the gastric cancer patients with stage II, and oxaliplatin combination therapy is recommended for the gastric cancer patients with stage III in Japan[2, 17–19]. Our study might suggest that

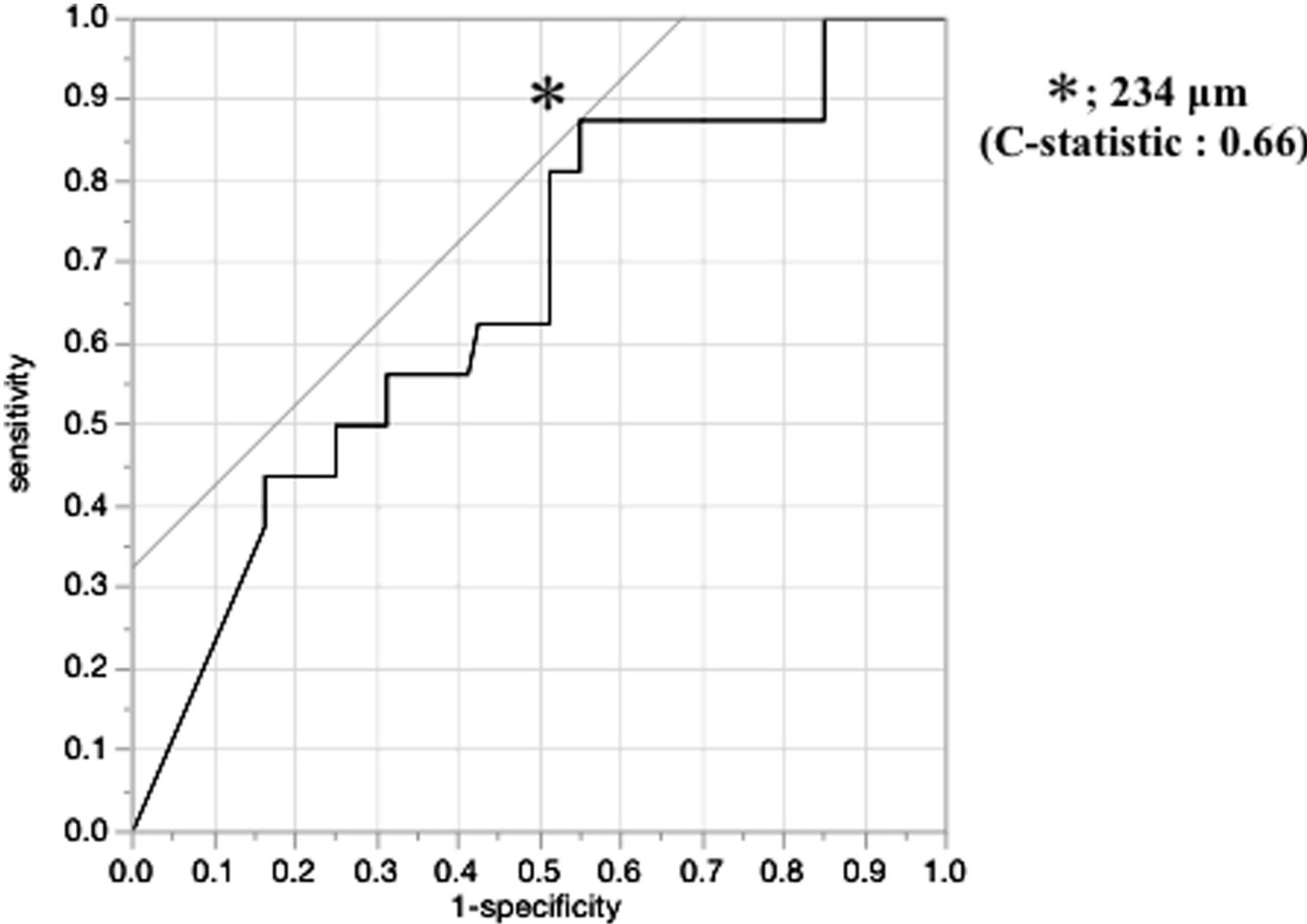

**Fig 3. Receiver operating characteristic (ROC) curve with the DIFS.** The cutoff value for DIFS was 234 μm.

oxaliplatin combination therapy may be recommended not only for stage III but also for stage II with DIFS ≤ 234 μm.

There were no significant factors associated with DIFS ≤234μm, but which tended to be associated with lymph node metastasis (p = 0.095; S1 Table). Lymph node metastasis was correlated with peritoneal recurrence in our study. It has been reported that peritoneal recurrence is caused by gastric cancer cells leave the primary tumor, adhere to the peritoneum, and proliferate at the site of adherence, resulting in the development of peritoneal recurrence[20, 21]. But some studies have suggested that peritoneal metastasis in gastric cancer without serosal invasion may occur via trans-lymphatic pathway[13, 22]. As for T3-stage gastric cancer with DIFS ≤234μm, our study might suggest same hypothesis.

It has been reported that E-cadherin, a cell-cell adhesion molecule, plays an important role in tumor invasion and distant metastasis such as peritoneal recurrence[12, 23, 24]. However, in the present study, no significant correlation was found between E-cadherin expression and peritoneal recurrence in gastric cancer at stage T3. One of the reasons why E-cadherin did not affect the peritoneal recurrence of T3 cases might be that E-cadherin might affect the invasion activity of cancer cells at early T-stage such as T1 and T2, but not advanced T-stage such as T3 and T4.

## E-cadherin positive

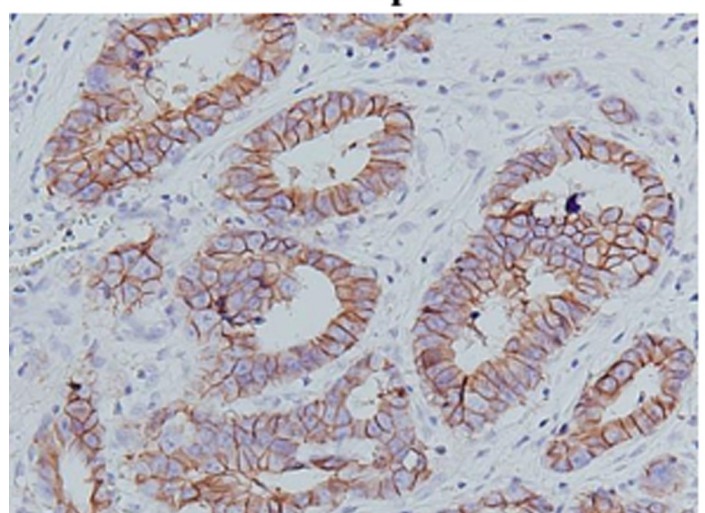

## E-cadherin negative

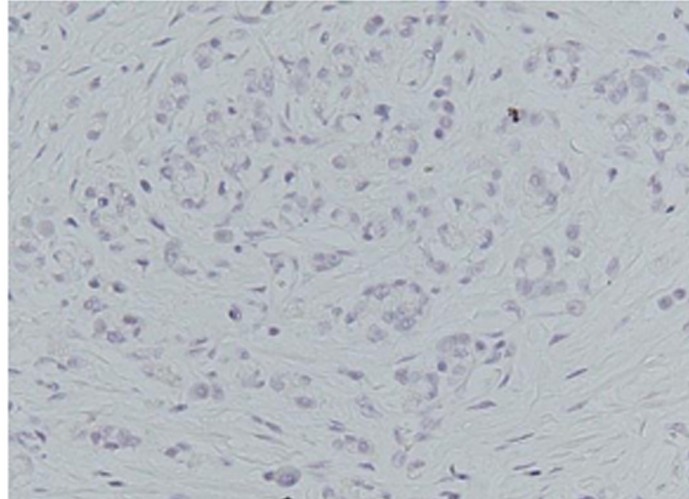

**Fig 4. E-cadherin staining.** E-cadherin was expressed mainly at the cell membrane.

In addition to H&E staining, pan-cytokeratin staining was used to evaluate the cancer cells at the invasion front. Pan-cytokeratin was expressed mainly in the cell membrane of cancer cells. The combination of pan-cytokeratin staining and H&E staining was a useful method to determine the cancer cells at the invasion front, especially for undifferentiated tumors. Pan-cytokeratin staining, which stains epithelial elements, makes the invasion front of cancer cells clear.

**Table 2. Univariate and multivariate analysis with respect to peritoneal recurrence.**

| Variables | Univariate analysis | | | Multivariate analysis | | |
|---|---|---|---|---|---|---|
| | Odds ratio | 95% CI | p-value | Odds ratio | 95% CI | p-value |
| E-cadherin | | | | | | |
| positive vs negative | 2.051 | 0.681–6.178 | 0.202 | | | |
| DIFS[a] | | | | | | |
| > 234 μm vs ≤ 234 μm | 5.727 | 1.221–26.868 | 0.027 | 4.862 | 1.005–23.516 | 0.049 |
| Macroscopic type | | | | | | |
| type1-2 vs type3-4 | 0.813 | 0.275–2.408 | 0.709 | | | |
| Histological type | | | | | | |
| intestinal vs diffuse | 1.229 | 0.415–3.639 | 0.710 | | | |
| LN metastasis[b] | | | | | | |
| negative vs positive | 5.571 | 1.472–21.083 | 0.011 | 4.846 | 1.249–18.803 | 0.023 |
| Lymphatic invasion | | | | | | |
| negative vs positive | 3.000 | 0.632–14.232 | 0.167 | | | |
| Vascular invasion | | | | | | |
| negative vs positive | 0.429 | 0.090–2.051 | 0.429 | | | |
| Tumor size | | | | | | |
| < 50 mm vs ≥ 50 mm | 1.421 | 0.482–4.188 | 0.524 | | | |

[a]: DIFS; the microscopic distance from tumor invasion front to serosa

[b]: LN metastasis; Lymph node metastasis

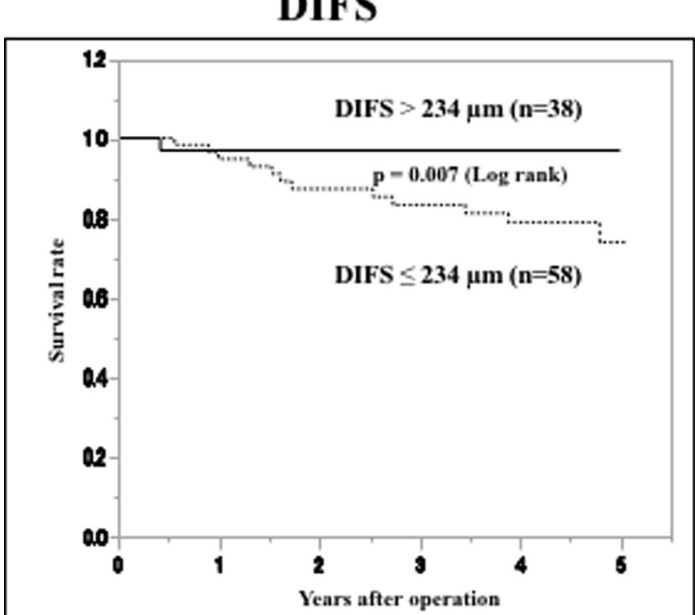
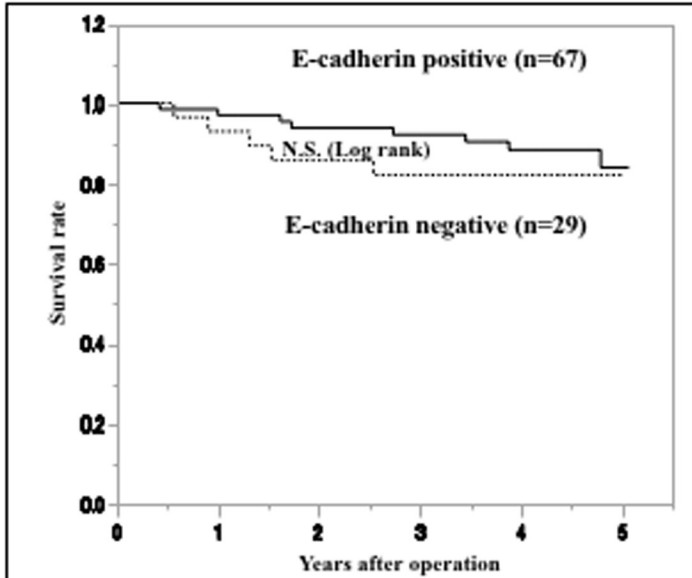

**Fig 5. Survival of the patients with gastric cancer.** The five-year overall survival of all patients (n = 96) based on the DIFS and on the E-cadherin expression. The Kaplan-Meier survival curve indicates that the five-year overall survival of the patients with a DIFS ≤234 μm was significantly worse than that of the patients with a DIFS >234 (p = 0.007). E-cadherin expression was not associated with the prognosis.

There may be limitations in this study. Since it is difficult to examine the whole lesions of tumor, it is uncertain that the obtained section represented most invasive lesion of cancer cells.

**Table 3. Univariate and multivariate analysis with respect to five-year overall survival.**

| Variables | Univariate analysis | | | Multivariate analysis | | |
|---|---|---|---|---|---|---|
| | Hazard ratio | 95% CI | p-value | Hazard ratio | 95% CI | p-value |
| E-cadherin | | | | | | |
| positive vs negative | 1.260 | 0.387–3.649 | 0.683 | | | |
| DIFS[a] | | | | | | |
| > 234 μm vs ≤ 234 μm | 9.834 | 1.955–178.670 | 0.003 | 8.752 | 1.670–160.900 | 0.005 |
| Macroscopic type | | | | | | |
| type1-2 vs type3-4 | 1.245 | 0.430–4.051 | 0.692 | | | |
| Histological type | | | | | | |
| intestinal vs diffuse | 0.884 | 0.272–2.559 | 0.884 | | | |
| LN metastasis[b] | | | | | | |
| negative vs positive | 4.186 | 1.306–18.514 | 0.015 | 3.582 | 1.091–16.104 | 0.032 |
| Lymphatic invasion | | | | | | |
| negative vs positive | 2.516 | 0.685–16.180 | 0.181 | | | |
| Vascular invasion | | | | | | |
| negative vs positive | 0.941 | 0.213–3.018 | 0.925 | | | |
| Tumor size | | | | | | |
| <50 mm vs ≥50 mm | 1.046 | 0.358–3.054 | 0.934 | | | |

[a]: DIFS; the microscopic distance from tumor invasion front to serosa

[b]: LN metastasis; Lymph node metastasis

In conclusion, the measurement of the DIFS might be useful for the prediction of peritoneal recurrence among gastric cancer patients who have undergone R0 curative surgery.

## Supporting information

**S1 Table. Correlation between peritoneal recurrence and clinicopathologic features in 96 gastric cancer cases at T3 stage.**
(DOCX)

**S2 Table. Correlation between DIFS ≤234 and lymph node metastasis.**
(DOCX)

## Acknowledgments

We thank Kayo Tsubota (Osaka City University Graduate School of Medicine), for technical assistance.

## Author Contributions

**Conceptualization:** Shingo Togano, Masakazu Yashiro.

**Data curation:** Yuichiro Miki, Sadaaki Nishimura.

**Formal analysis:** Yurie Yamamato, Mami Yoshii, Hiroaki Tanaka.

**Funding acquisition:** Tomohiro Sera, Tomohisa Okuno.

**Investigation:** Yukako Kushitani, Atsushi Sugimoto, Tatsuro Tamura, Kazuya Muguruma, Sayaka Tanaka.

**Methodology:** Shuhei Kushiyama, Kenji Kuroda.

**Project administration:** Masaichi Ohira.

**Resources:** Takahiro Toyokawa.

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
