## [Decision Letter · Decision Letter 0]

30 Sep 2019

PONE-D-19-23016

Microscopic distance from tumor invasion front to serosa might be a useful predictive factor for peritoneal recurrence after curative resection of gastric cancer

PLOS ONE

Dear Dr. Yashiro,

Thank you for submitting your manuscript to PLOS ONE. After careful consideration, we feel that it has merit but does not fully meet PLOS ONE’s publication criteria as it currently stands. Therefore, we invite you to submit a revised version of the manuscript that addresses the points raised during the review process.

We would appreciate receiving your revised manuscript by Nov 14 2019 11:59PM. To enhance the reproducibility of your results, we recommend that if applicable you deposit your laboratory protocols in protocols.io, where a protocol can be assigned its own identifier (DOI) such that it can be cited independently in the future. For instructions see: http://journals.plos.org/plosone/s/submission-guidelines#loc-laboratory-protocols

We look forward to receiving your revised manuscript.

Kind regards,

Kun Yang

Academic Editor

PLOS ONE

Journal Requirements:

https://www.sciencedirect.com/science/article/pii/S0960740417302025?via%3Dihub

The text that needs to be addressed is in the Introduction section.

In your revision ensure you cite all your sources (including your own works), and quote or rephrase any duplicated text outside the methods section. Further consideration is dependent on these concerns being addressed.

'This study is partially founded by KAKENHI Grant-in-Aid for Scientific Research, Nos. 18H02883(M.Y.).'

'KAKENHI Grant-in-Aid for Scientific Research, Nos. 18H02883(M.Y.).'

Additional Editor Comments (if provided):

Please see the comments below

Reviewers' comments:

Reviewer's Responses to Questions

**Comments to the Author**

1. Is the manuscript technically sound, and do the data support the conclusions?

Reviewer #1: Partly

Reviewer #2: Yes

Reviewer #3: Yes

Reviewer #4: Yes

2. Has the statistical analysis been performed appropriately and rigorously? 

Reviewer #1: No

Reviewer #2: Yes

Reviewer #3: Yes

Reviewer #4: Yes

3. Have the authors made all data underlying the findings in their manuscript fully available?

Reviewer #1: Yes

Reviewer #2: Yes

Reviewer #3: Yes

Reviewer #4: Yes

4. Is the manuscript presented in an intelligible fashion and written in standard English?

Reviewer #1: Yes

Reviewer #2: Yes

Reviewer #3: Yes

Reviewer #4: Yes

5. Review Comments to the Author

Reviewer #1: The authors reported the predictive significance of microscopic distance from tumor invasion front to serosa for peritoneal recurrence in T3-stage gastric cancer. The results of this study are interesting, but I think it is natural that the deeper depth of invasion was associated with the more frequent peritoneal recurrence. In addition, authors should revise following several points to improve the content.

1. This study evaluated only T3-gastric cancer.I suggest to change the title.

"gastric cancer" to "T3-gastric cancer"

2. Why did the authors focus on the microscopic distance from the tumor invasion front to the serosa (DIFS) and E-cadherin. Please mentioned in Introduction section.

3. Please clarify the inclusion criteria. The authors should describe the study period and flow chart.

4. The authors have used the UICC/AJCC 7th edition staging manual. Please use the most current staging manual.

5. As the authors mentioned in Introduction section, we sometimes experienced peritoneal recurrence even in T1-2 cancer patients.Why T1-2 cancer was not included in this analysis? They should included T1-2 cancers in this study.

6. In this analysis, location of the tumor (Upper/Middle/Lower and Anterior/Posterior/Grater curvature side/Lesser curvature side) was not included. These factor should be included.

Reviewer #2: The authors demonstrated that short distance from the tumor invasion front to the serosa (DIFS) was an independent risk factor for peritoneal recurrence and unfavorable survival. This study may have considerable clinical implications, but there are several problems in their presentation that need to be solved:

1. Did the authors examine peritoneal washing cytology? If so, the authors should describe the result and analyze the relationship between peritoneal recurrence and cytology.

2. The authors should show the clinicopathological features in more detail in Table 1, including tumor location, type of gastrectomy, extent of lymph node dissection, and the number of the patients who underwent adjuvant chemotherapy.

3. In Material and methods section (page 5, line 77-82), the author should describe who evaluated E-cadherin IHC staining (e.g. by experienced pathologist unaware of clinical data). If possible, the authors should perform the agreement study of E-cadherin expression and show the concordance between the two pathologist.

4. The authors should discuss several limitations of this study in Discussion section.

5. Please recheck the grammar and terminology.

6. The statistical method for the multivariate analysis is unclear.

Reviewer #3: Togano, et al. investigated the risk factors of peritoneal recurrence in patients who underwent curative gastrectomy for T3 gastric cancer. They evaluated the correlation between the occurrence of peritoneal recurrence and clinicopathologic factors including distance from tumor invasion front to serosa (DIFS) and expression status of E-Cadherin, and found that lymph node metastasis and DIFS were the independent factors associated with peritoneal recurrence. This manuscript will provide very important clinical information for readers. Several changes will improve the quality of this manuscript.

1. The authors focused on the risk factors of peritoneal metastasis after curative resection for T3 gastric cancer. The title does not include the information that this study especially focused on T3 gastric cancer.

2. The follow-up period of the study cohort is very important information for readers. Besides, it is also important how these patients were followed up. The authors should provide the information on the follow-up period, timing and modality for follow-up.

3. Adjuvant chemotherapy may influence the survival of patients. Please include the information on presence or absence of adjuvant chemotherapy into the background.

4. When discussing the DIFS, the preparation of tissue section for pathology is very important information. Please provide the details of preparation for tissue section.

5. Please provide the C-statistic of the ROC curve shown in Figure 2.

6. The authors should describe the detail of multivariate analysis. How did they select the covariates?

7. The discussion seems to be poor. Please discuss why the expression of E-Cadherin did not influence the peritoneal metastasis in this study cohort, and the clinical evidence of adjuvant therapy to decrease peritoneal recurrence. Also, they should declare the limitation of this study.

Reviewer #4: Comments for the author

Decision: Minor revision

This article identified a predictive factor for peritoneal recurrence of T3 gastric cancer patients. They focused on the distance from the tumor invasion front to the serosa (DIFS) and clarified the cut off value of DIFS for the prediction of peritoneal recurrence.

Major points

1. DIFS is very important in this article. So they should describe the precise method to measure the DIFS. For example, how many did they check to confirm the invasion front. They should clarify the identification of DIFS in details.

2. They classified the patients to peritoneal recurrence and non-recurrence groups. How long did they observe those patients? They should describe the observation period.

3. The explanation of the importance of E-cadherin in gastric cancer patients is necessary.

4. DIFS and lymph node metastasis are independent predictive factors for peritoneal recurrence. So they should add subgroup analysis in T3, lymph node metastasis negative patients.

Minor points

1. In Table1, they should include DIFS and E-cadherin expression.

2. In line 189, “by H&E staining” is unnecessary.

6. PLOS authors have the option to publish the peer review history of their article (what does this mean?). If published, this will include your full peer review and any attached files.

Reviewer #1: No

Reviewer #2: No

Reviewer #3: Yes: Masayuki Watanabe

Reviewer #4: Yes: Shinichiro Hasegawa

---

## [Author Response · Author response to Decision Letter 0]

31 Oct 2019

Dear Kun Yang Ph.D.

Academic Editor 

We greatly appreciate your invitation for us to revise our article “Microscopic distance from tumor invasion front to serosa might be a useful predictive factor for peritoneal recurrence after curative resection of T3-gastric cancer”. We would like to thank you for a number of comments and suggestions for improvement in our manuscript. We have carefully considered the referees’ comments and have made point-by-point responses as described below. Also, we highlight all changes in the revised manuscript. This manuscript is not being considered in whole or in part by any other journal. All authors are aware of the content of this manuscript.

We hope you will seriously consider this report for publication in PLOS ONE.

Dear Reviewer#1

Thank you very much for the careful review of the Reviewer #1. We corrected several points according to the descriptions by the Reviewer #1, as described below. We indicated the changes point by point and highlighted them in the revised paper.

1. This study evaluated only T3-gastric cancer.

I suggest to change the title. "gastric cancer" to "T3-gastric cancer"

According to the reviewer#1's comment, we change the title, "Microscopic distance from tumor invasion front to serosa might be a useful predictive factor for peritoneal recurrence after curative resection of T3-gastric caner". (on page 1, line 3)

2. Why did the authors focus on the microscopic distance from the tumor invasion front to the serosa (DIFS) and E-cadherin. Please mentioned in Introduction section.

It has been reported that E-cadherin is one of important factors for tumor invasion and distant metastasis in some solid cancers [8, 10-12]. Taken together, we previously reported the correlation between DIFS and serosal exposure of gastric cancer cells, and speculated that DIFS might be associated with peritoneal recurrence [3]. Then, in this study we focused on the significance of DIFS and E-cadherin in peritoneal recurrence. We added the comments in the introduction. (on page 3 line 40-45)

[3] Miki Y, Yashiro M, Ando K, Okuno T, Kitayama K, Masuda G, et al. Examination of cancer cells exposed to gastric serosa by serosal stamp cytology plus RT-PCR is useful for the identification of gastric cancer patients at high risk of peritoneal recurrence. Surgical oncology. 2017;26(4):352-8. Epub 2017/11/09. doi: 10.1016/j.suronc.2017.07.008. PubMed PMID: 29113652.

[8] Sun F, Feng M, Guan W. Mechanisms of peritoneal dissemination in gastric cancer. Oncology letters. 2017;14(6):6991-8. Epub 2018/01/19. doi: 10.3892/ol.2017.7149. PubMed PMID: 29344127; PubMed Central PMCID: PMCPMC5754894.

[10] Gao M, Zhang X, Li D, He P, Tian W, Zeng B. Expression analysis and clinical significance of eIF4E, VEGF-C, E-cadherin and MMP-2 in colorectal adenocarcinoma. Oncotarget. 2016;7(51):85502-14. Epub 2016/12/03. doi: 10.18632/oncotarget.13453. PubMed PMID: 27907907; PubMed Central PMCID: PMCPMC5356753.

[11] Liang G, Ding M, Lu H, Cao NA, Niu Y, Gao Y, et al. Metformin upregulates E-cadherin and inhibits B16F10 cell motility, invasion and migration. Oncology letters. 2015;10(3):1527-32. Epub 2015/12/02. doi: 10.3892/ol.2015.3475. PubMed PMID: 26622703; PubMed Central PMCID: PMCPMC4533732.

[12] Torabizadeh Z, Nosrati A, Sajadi Saravi SN, Yazdani Charati J, Janbabai G. Evaluation of E-cadherin Expression in Gastric Cancer and Its Correlation with Clinicopathologic Parameters. International journal of hematology-oncology and stem cell research. 2017;11(2):158-64. Epub 2017/09/07. PubMed PMID: 28875011; PubMed Central PMCID: PMCPMC5575728.

3. Please clarify the inclusion criteria. The authors should describe the study period and flow chart.

The inclusion criteria were as follows; 1. histologically proven gastric adenocarcinoma; 2. the depth of tumor invasion was T3; 3. curative operation; 4. intraoperative peritoneal lavage cytology-negative. The follow-up period was 60 months, and the median follow-up was 49.3 months. We described these criteria in the materials and Fig. 1. (on page 4 line 52-58)

4. The authors have used the UICC/AJCC 7th edition staging manual. Please use the most current staging manual.

We changed the 7th edition of TNM Classification to the 8th edition of TNM Classification [15]. (on page 5 line 71)

[15] Brierley JD, Gospodarowicz MK, Writtekind C. TNM Classification of Malignant Tumours, 8th Edition 2016.

5. As the authors mentioned in Introduction section, we sometimes experienced peritoneal recurrence even in T1-2 cancer patients. Why T1-2 cancer was not included in this analysis? They should include T1-2 cancers in this study.

Since the peritoneal recurrence of T1 and T2 cancers has been considered to develop via trans-lymphatic pathway [13, 14], we excluded T1 and T2 cases in this study. (on page 4 line 55-57)

[13] Yoshida M, Sugino T, Kusafuka K, Nakajima T, Makuuchi R, Tokunaga M, et al. Peritoneal dissemination in early gastric cancer: importance of the lymphatic route. Virchows Arch. 2016;469(2):155-61. Epub 2016/05/26. doi: 10.1007/s00428-016-1960-7. PubMed PMID: 27220762.

[14] Yamamoto M, Taguchi K, Baba H, Endo K, Kohnoe S, Okamura T, et al. Peritoneal dissemination of early gastric cancer: report of a case. Surgery today. 2006;36(9):835-8. Epub 2006/08/29.

6. In this analysis, location of the tumor (Upper/ Middle/ Lower and Anterior/ Posterior/ Grater curvature side/ Lesser curvature side) was not included. These factors should be included.

We included the tumor location in S1 Table.

Dear Reviewer#2

Thank you very much for the careful review of the Reviewer #2. We correct several points according to the descriptions by the Reviewer #2, as follows.

1. Did the authors examine peritoneal washing cytology? If so, the authors should describe the result and analyze the relationship between peritoneal recurrence and cytology.

We performed peritoneal washing cytology in all cases. Peritoneal washing cytology-positive cases were excluded in this study, which was described in the materials and Fig. 1. (on page 4 line 52-55)

2. The authors should show the clinicopathological features in more detail in Table 1, including tumor location, type of gastrectomy, extent of lymph node dissection, and the number of the patients who underwent adjuvant chemotherapy.

We added the detail of clinicopathological features, including tumor location, type of gastrectomy, extent of lymph node dissection, and the number of the patients who underwent adjuvant chemotherapy in S1 Table.

3. In Material and methods section (page 5, line 77-82), the author should describe who evaluated E-cadherin IHC staining (e.g. by experienced pathologist unaware of clinical data). If possible, the authors should perform the agreement study of E-cadherin expression and show the concordance between the two pathologist.

The pathologist, Dr Sayaka Tanaka, checked DIFS and E-cadherin expression. Dr Togano S, Dr Yashiro M, and Dr Tanaka S checked DIFS and E-cadherin expression. We added these comments in the materials and methods. (on page 6 line 119-121)

4. The authors should discuss several limitations of this study in Discussion section.

We added limitations in the manuscript, as follows. Since it is difficult to examine the whole lesions of tumor, it is uncertain that the obtained section represented most invasive lesion of cancer cells. (on page 14 line 347-349)

5. Please recheck the grammar and terminology.

The manuscript was proofread by a professional editor and native English speaker.

6. The statistical method for the multivariate analysis is unclear.

Multivariate analysis with respect to peritoneal recurrence was performed using logistic regression analysis. Multivariate analysis with respect to five-year overall survival was performed using Cox proportional hazard model. Covariates were selected from those with significant differences in univariate analysis. (on page 7 line 130-133)

Dear Reviewer#3

Thank you very much for the careful review of the Reviewer #3. We correct several points according to the descriptions by the Reviewer #3, as follows.

1. The authors focused on the risk factors of peritoneal metastasis after curative resection for T3 gastric cancer. The title does not include the information that this study especially focused on T3 gastric cancer.

According to the reviewer's comments, we changed the title, "Microscopic distance from tumor invasion front to serosa might be a useful predictive factor for peritoneal recurrence after curative resection of T3-gastric cancer" (on page 1, line 3)

2. The follow-up period of the study cohort is very important information for readers. Besides, it is also important how these patients were followed up. The authors should provide the information on the follow-up period, timing and modality for follow-up.

We added follow-up period, timing and modality in Materials and Method section, as follows. The follow-up period was 60 months, and the median follow-up was 49.3 months. The follow-up program of postoperative surveillance consisted of computed tomography, and ultrasound performed every 3 months in order to diagnose recurrent diseases. (on page 4 line 57-60)

3. Adjuvant chemotherapy may influence the survival of patients. Please include the information on presence or absence of adjuvant chemotherapy into the background.

We included the data of adjuvant chemotherapy in S1 Table.

4. When discussing the DIFS, the preparation of tissue section for pathology is very important information. Please provide the details of preparation for tissue section.

The preparation of tissue section was as follows. After gastrectomy, the gastric tumor was immediately treated with 10% formalin neutral buffer solution for 24-72 hours. We added the comments in Materials and Method section. (on page 5 line 79-80)

5. Please provide the C-statistic of the ROC curve shown in Figure 2.

We added the C-statistic (0.66) in Figure 2.

6. The authors should describe the detail of multivariate analysis. How did they select the covariates?

Multivariate analysis with respect to peritoneal recurrence was performed using logistic regression analysis. Multivariate analysis with respect to five-year overall survival was performed using Cox proportional hazard model. Covariates were selected from those with significant differences in univariate analysis. (on page 7 line 130-133)

7. The discussion seems to be poor. Please discuss why the expression of E-Cadherin did not influence the peritoneal metastasis in this study cohort, and the clinical evidence of adjuvant therapy to decrease peritoneal recurrence. Also, they should declare the limitation of this study.

One of the reasons why E-cadherin did not affect the peritoneal recurrence of T3 cases might be that E-cadherin might affect the invasion activity of cancer cells at early T-stage such as T1 and T2, but not advanced T-stage such as T3 and T4. (on page 13 line 334- page 13 line 340)

According to guideline, tegafur-gimeracil-oteracil (TS-1) monotherapy is recommended for the gastric cancer patients with stage II, and oxaliplatin combination therapy is recommended for the gastric cancer patients with stage III in Japan [2.17-19]. Our study might suggest that oxaliplatin combination therapy may be recommended not only for stage III but also for stage II with DIFS ≤ 234 μm. (on page 13 line 317-321).

We added limitations in the manuscript, as follows. Since it is difficult to examine the whole lesions of tumor, it is uncertain that the obtained section represented most invasive lesion of cancer cells. (on page 14 line 348-350).

[2] Sakuramoto S, Sasako M, Yamaguchi T, Kinoshita T, Fujii M, Nashimoto A, et al. Adjuvant chemotherapy for gastric cancer with S-1, an oral fluoropyrimidine. The New England journal of medicine. 2007;357(18):1810-20. Epub 2007/11/06. doi: 10.1056/NEJMoa072252. PubMed PMID: 17978289. 

[17] Sasako M, Sakuramoto S, Katai H, Kinoshita T, Furukawa H, Yamaguchi T, et al. Five-year outcomes of a randomized phase III trial comparing adjuvant chemotherapy with S-1 versus surgery alone in stage II or III gastric cancer. Journal of clinical oncology : official journal of the American Society of Clinical Oncology. 2011;29(33):4387-93. Epub 2011/10/20. doi: 10.1200/jco.2011.36.5908. PubMed PMID: 22010012. 

[18] Bang YJ, Kim YW, Yang HK, Chung HC, Park YK, Lee KH, et al. Adjuvant capecitabine and oxaliplatin for gastric cancer after D2 gastrectomy (CLASSIC): a phase 3 open-label, randomised controlled trial. Lancet (London, England). 2012;379(9813):315-21. Epub 2012/01/10. doi: 10.1016/s0140-6736(11)61873-4. PubMed PMID: 22226517. 

[19] Noh SH, Park SR, Yang HK, Chung HC, Chung IJ, Kim SW, et al. Adjuvant capecitabine plus oxaliplatin for gastric cancer after D2 gastrectomy (CLASSIC): 5-year follow-up of an open-label, randomised phase 3 trial. The Lancet Oncology. 2014;15(12):1389-96. Epub 2014/12/03. doi: 10.1016/s1470-2045(14)70473-5. PubMed PMID: 25439693. 

Dear Reviewer#4

Thank you very much for the careful review of the Reviewer #4. We correct several points according to the descriptions by the Reviewer #4, as follows.

Major points

1. DIFS is very important in this article. So they should describe the precise method to measure the DIFS. For example, how many did they check to confirm the invasion front. They should clarify the identification of DIFS in details.

We checked invasion depth of all specimens of the tumor using H-E staining slides. After selecting three specimens with high invasion depth, the three specimens were stained by Pan-cytokeratin and measured DIFS of 3 slides. The shortest distance was defined as DIFS of the case. We added the comments in Materials and Method section. (on page 6 line 108-111)

2. They classified the patients to peritoneal recurrence and non-recurrence groups. How long did they observe those patients? They should describe the observation period.

The follow-up period was 60 months, and the median follow-up was 49.3 months. We added follow-up period in Materials and Method section. (on page 4 line 57-58)

3. The explanation of the importance of E-cadherin in gastric cancer patients is necessary.

It has been reported that E-cadherin is one of important factors for tumor invasion and distant metastasis in some solid cancers [8, 10-12]. We added the comments in the introduction. (on page 3 line 40-45)

[8] Sun F, Feng M, Guan W. Mechanisms of peritoneal dissemination in gastric cancer. Oncology letters. 2017;14(6):6991-8. Epub 2018/01/19. doi: 10.3892/ol.2017.7149. PubMed PMID: 29344127; PubMed Central PMCID: PMCPMC5754894.

[10] Gao M, Zhang X, Li D, He P, Tian W, Zeng B. Expression analysis and clinical significance of eIF4E, VEGF-C, E-cadherin and MMP-2 in colorectal adenocarcinoma. Oncotarget. 2016;7(51):85502-14. Epub 2016/12/03. doi: 10.18632/oncotarget.13453. PubMed PMID: 27907907; PubMed Central PMCID: PMCPMC5356753.

[11] Liang G, Ding M, Lu H, Cao NA, Niu Y, Gao Y, et al. Metformin upregulates E-cadherin and inhibits B16F10 cell motility, invasion and migration. Oncology letters. 2015;10(3):1527-32. Epub 2015/12/02. doi: 10.3892/ol.2015.3475. PubMed PMID: 26622703; PubMed Central PMCID: PMCPMC4533732.

[12] Torabizadeh Z, Nosrati A, Sajadi Saravi SN, Yazdani Charati J, Janbabai G. Evaluation of E-cadherin Expression in Gastric Cancer and Its Correlation with Clinicopathologic Parameters. International journal of hematology-oncology and stem cell research. 2017;11(2):158-64. Epub 2017/09/07. PubMed PMID: 28875011; PubMed Central PMCID: PMCPMC5575728.

4. DIFS and lymph node metastasis are independent predictive factors for peritoneal recurrence. So they should add subgroup analysis in T3, lymph node metastasis negative patients.

We performed subgroup analysis for lymph node metastasis negative patients (n=48), positive patients (n=48). Only 3 cases developed in lymph node metastasis negative patients (n=48). There was no significant correlation between DIFS ≤234 μm and peritoneal recurrence (data not shown).

Minor points

1. In Table1, they should include DIFS and E-cadherin expression.

2. In line 189, “by H&E staining” is unnecessary.

According to the comment, we include DIFS and E-cadherin expression to Table 1 and omit a sentence, “by H&E staining” In line 189.

Dear Reviewer#5

Thank you very much for the careful review of the Reviewer #5. We correct several points according to the descriptions by the Reviewer #5, as follows.

1. The authors measured the microscopic distance from the tumor invasion front to the serosa (DIFS) using tissue slides by H&E staining and pan-cytokeratin. However, there are no description about the numbers of slides they evaluated the DIFS. It might be necessary to measure at least 3 slides per case to evaluate DIFS precisely.

We checked invasion depth of all specimens of the tumor using H-E staining slides. After selecting three specimens with high invasion depth, the three specimens were stained by Pan-cytokeratin and measured DIFS of 3 slides. The shortest distance was defined as DIFS of the case. We added the comments in Materials and Method section. (on page 6 line 108-111)

2. The authors did not mention about the adjuvant chemotherapy. The authors should describe the presence or absence of adjuvant chemotherapy, and also indicate the contents of adjuvant chemotherapy.

We showed the data of adjuvant chemotherapy in S1 Table.

3. The authors pointed out that the lymph node metastasis were independent risk factors for peritoneal recurrence. And they assumed that the serosal invasion may occur via translymphatic pathway. It should be desireble to present the mechanism of serosal invasion may occur via translymphatic pathway. And they should evaluate the exposure of cancer cell at the surface of lymph node which the cancer cells metastases.

We commented as follows; peritoneal recurrence in gastric cancer without serosal invasion may occur via trans-lymphatic pathway. (on page 13 line 327-330)

Evaluation of the exposure of cancer cells from the lymph nodes may be useful to prove peritoneal recurrence via trans-lymphatic pathway. However, lymph node metastasis might be secondary developed after peritoneal recurrence. Therefore, it is difficult to evaluate whether peritoneal metastasis might occur via trans-lymphatic pathway or serosal exposure of cancer cells.

---

## [Decision Letter · Decision Letter 1]

18 Nov 2019

Microscopic distance from tumor invasion front to serosa might be a useful predictive factor for peritoneal recurrence after curative resection of T3-gastric cancer

PONE-D-19-23016R1

Dear Dr. Yashiro,

We are pleased to inform you that your manuscript has been judged scientifically suitable for publication and will be formally accepted for publication once it complies with all outstanding technical requirements.

With kind regards,

Kun Yang

Academic Editor

PLOS ONE

Additional Editor Comments (optional):

Please change from "TS-1" to "S-1".

Reviewers' comments:

Reviewer's Responses to Questions

**Comments to the Author**

1. If the authors have adequately addressed your comments raised in a previous round of review and you feel that this manuscript is now acceptable for publication, you may indicate that here to bypass the “Comments to the Author” section, enter your conflict of interest statement in the “Confidential to Editor” section, and submit your "Accept" recommendation.

Reviewer #1: All comments have been addressed

Reviewer #2: All comments have been addressed

Reviewer #3: All comments have been addressed

Reviewer #4: (No Response)

2. Is the manuscript technically sound, and do the data support the conclusions?

Reviewer #1: Yes

Reviewer #2: Yes

Reviewer #3: Yes

Reviewer #4: (No Response)

3. Has the statistical analysis been performed appropriately and rigorously? 

Reviewer #1: Yes

Reviewer #2: Yes

Reviewer #3: Yes

Reviewer #4: (No Response)

4. Have the authors made all data underlying the findings in their manuscript fully available?

Reviewer #1: Yes

Reviewer #2: Yes

Reviewer #3: Yes

Reviewer #4: (No Response)

5. Is the manuscript presented in an intelligible fashion and written in standard English?

Reviewer #1: Yes

Reviewer #2: Yes

Reviewer #3: Yes

Reviewer #4: (No Response)

6. Review Comments to the Author

Reviewer #1: The authors responded well to the comments and the manuscript was improved substantially. However, authors should revise the following point.

1. Change from "TS-1" to "S-1".

Reviewer #2: I think that the authors have answered appropriately reviewers’ comments point-by-point and this article is now acceptable for the PLOS ONE.

Reviewer #3: Togano, et al. investigated the risk factors of peritoneal recurrence in patients who underwent curative gastrectomy for T3 gastric cancer. They evaluated the correlation between the occurrence of peritoneal recurrence and clinicopathologic factors including distance from tumor invasion front to serosa (DIFS) and expression status of E-Cadherin, and found that lymph node metastasis and DIFS were the independent factors associated with peritoneal recurrence. The authors appropriately revised the manuscript.

Reviewer #4: (No Response)

7. PLOS authors have the option to publish the peer review history of their article (what does this mean?). If published, this will include your full peer review and any attached files.

Reviewer #1: No

Reviewer #2: No

Reviewer #3: Yes: Masayuki Watanabe

Reviewer #4: No

---

## [Editor Report · Acceptance letter]

26 Nov 2019

PONE-D-19-23016R1 

Microscopic distance from tumor invasion front to serosa might be a useful predictive factor for peritoneal recurrence after curative resection of T3-gastric cancer 

Dear Dr. Yashiro:

I am pleased to inform you that your manuscript has been deemed suitable for publication in PLOS ONE. Congratulations! Your manuscript is now with our production department. 

With kind regards,

on behalf of

Dr. Kun Yang 

Academic Editor

PLOS ONE